# Understanding the PEDOT:PSS, PTAA and P3CT-X Hole-Transport-Layer-Based Inverted Perovskite Solar Cells

**DOI:** 10.3390/polym14040823

**Published:** 2022-02-21

**Authors:** Qi Bin Ke, Jia-Ren Wu, Chia-Chen Lin, Sheng Hsiung Chang

**Affiliations:** 1Department of Physics, Chung Yuan Christian University, Taoyuan 320314, Taiwan; qibin050751@gmail.com (Q.B.K.); jrwu@cycu.org.tw (J.-R.W.); s10712114@cycu.org.tw (C.-C.L.); 2R&D Center for Membrane Technology and Center for Nano Technology, Chung Yuan Christian University, Taoyuan 320314, Taiwan

**Keywords:** p-type polymers, inverted perovskite solar cells, nucleation, crystal growth, perovskite thin film, interfacial contacts

## Abstract

The power conversion efficiencies (PCEs) of metal-oxide-based regular perovskite solar cells have been higher than 25% for more than 2 years. Up to now, the PCEs of polymer-based inverted perovskite solar cells are widely lower than 23%. PEDOT:PSS thin films, modified PTAA thin films and P3CT thin films are widely used as the hole transport layer or hole modification layer of the highlyefficient inverted perovskite solar cells. Compared with regular perovskite solar cells, polymer-based inverted perovskite solar cells can be fabricated under relatively low temperatures. However, the intrinsic characteristics of carrier transportation in the two types of solar cells are different, which limits the photovoltaic performance of inverted perovskite solar cells. Thanks to the low activation energies for the formation of high-quality perovskite crystalline thin films, it is possible to manipulate the optoelectronic properties by controlling the crystal orientation with the different polymer-modified ITO/glass substrates. To achieve the higher PCE, the effects of polymer-modified ITO/glass substrates on the optoelectronic properties and the formation of perovskite crystalline thin films have to be completely understood simultaneously.

## 1. Introduction

Conductive polymers are widely used in organic photovoltaics (OPVs) and dye-sensitized solar cells (DSSCs) as the hole transport layer (HTL) due to their high transparency, large work function and high carrier mobility [1,2,3,4,5,6]. Poly(3,4-ethylenedioxythiophene)polystyrene sulfonate (PEDOT:PSS) and polyaniline thin films are the most commonly used p-type polymer in organic-related solar cells [7,8,9,10]. Ten years ago, the power conversion efficiencies (PCEs) of OPVs and DSSCs were lower than 10% mainly due to large potential loss [11,12] and strong exciton binding energy [13,14] of the active layer. Fortunately, the organic light-absorbing materials can be replaced by the high-quality perovskite crystalline thin films which can be fabricated by using various solution process techniques under low temperatures ranging from 60 °C to 140 °C [15,16,17,18,19], which have largely boosted the PCE of organic-based solar cells to be higher than 20% in the recent decade. The PEDOT:PSS thin film was used in the first inverted perovskite solar cells, which resulted in a moderate PCE of 3.9% [20]. The PEDOT:PSS thin films are deposited on top of the ITO/glass substrates as the HTL and electron-blocking layer (EBL), which can influence the open-circuit voltage (V_OC_), short-circuit current density (J_SC_) and fill factor (FF) of the resultant perovskite solar cells by varying the molecular structure of the PEDOT chains and the thickness of the PEDOT:PSS thin films [21,22,23,24]. However, the PCE values of PEDOT:PSS-based inverted perovskite solar cells are lower than 20%, mainly due to the relatively low V_OC_ and FF [25,26,27,28,29,30], which are originated from the potential loss at the perovskite/PEDOT:PSS interface. Poly(triarylamine) (PTAA)- and poly[3-(4-carboxybutyl) thiophene-2,5-diyl] (P3CT)-based thin films have been widely used to replace the PEDOT:PSS thin films, which can increase the V_OC_ and FF of the resultant perovskite solar cells simultaneously [31,32,33,34]. It can be explained as being due to the reductions of potential loss and carrier recombination in the perovskite layer and at the perovskite/HTL interface [35,36,37,38]. The surface wettabilities of the PEDOT:PSS, P3CT-X (X: Na, K, Rb, Cs) and PTAA thin films are super-hydrophilic, hydrophilic and hydrophobic, respectively, which can largely influence the film discontinuity and grain size of the deposited perovskite thin films [39,40].

Hydrophobic NiO_x_- and CuO_x_-based thin films are also widely used in the inverted perovskite solar cells as the HTL [41,42]. The photovoltaic performance of NiO_x_-based inverted perovskite solar cells can be increased to be higher than 20% by using an organic interlayer in between the perovskite thin film and the HTL [43,44,45,46,47]. In other words, the formation of high-quality perovskite crystalline thin films is not only related to the surface wettability [48,49,50] but is also dominated by the nucleation process of perovskites on top of the organic layers. Conceptually, the formation of uniform nucleation sites can form preferred oriented perovskite crystalline thin films, which is similar to the crystal growth of perovskites on top of the single-crystalline wafer [51] or organic self-assemble (SAM) monolayer modified substrates [52,53].

On the other hand, the surface properties of HTL dominate the grain size, surface roughness and crystal orientation of the resultant perovskite thin films and thereby influence the contact quality at the electron transport layer (ETL)/perovskite interface. In the inverted perovskite solar cells, C_60_ and (phenyl-C_61_-butyric acid methyl ester) PCBM thin films are widely used as the ETL [54,55,56,57]. It is noted that the surface roughness of perovskite crystalline thin films determines the formation of the s-shaped J-V curves in the PCBM/MAPbI_3_ heterojunction solar cells [58,59,60]. Besides, the solution-processed bathocuproine (BCP) and thermal evaporated BCP can be used to modify the C_60_-derivatives-based ETL, which largely increases the FF of the resultant perovskite solar cells [61,62,63,64]. BCP molecules and oxygen-containing functional group of PCBM molecules can passivate the electron-poor defects at the grain boundaries of perovskite thin films due to the sub-nanometer scale. C_60_ molecules can passivate the electron-rich defects at the grain boundaries of perovskite thin films due to the electron chargeable property. However, the V_OC_ hysteresis can still be observed in the J-V curves, which indicates that the surface defects of perovskite crystalline thin films are not completely passivated by the capping layer. In other words, the crystal orientation of perovskite thin films plays an important role, which determines the types of surface defects in the top region.

According to the theoretical calculations, the highest PCE of the inverted perovskite solar cells is about 30% [65,66], which is lower than the highest prediction value from Shockley–Queisser (S.-Q.) limit because the absorption bandgap of lead trihalide-based perovskite material is higher than the optimal absorption bandgap [67,68]. Up to now, the highest PCE values of regular perovskite solar cells and inverted perovskite solar cells are 25.59% [69] and 23.32% [70], respectively. In the best regular perovskite solar cell, a mesoporous-TiO_2_/compact-TiO_2_ bilayer structure is used as the ETL. In the best inverted perovskite solar cell, a phenylethylammonium iodide (PEAI)-modified PTAA thin film is used as the HTL while improving the contact quality at the ETL/perovskite interface via the treatment of PEAI molecules. Compared with the photovoltaic performance of the most regular perovskite solar cells, the V_OC_ and FF of the most inverted perovskite solar cells are relatively lower mainly due to the larger potential loss and higher carrier recombination. In other words, there is still room for improvement in the PCE of inverted perovskite solar cells.

In this review, we focus on the understanding of highly efficient inverted perovskite solar cells. In the following sections, the working mechanism of perovskite solar cells will be mentioned first. The research progress of the polymer-based inverted perovskite solar cells will be mentioned in order to discuss the possible future directions, which is divided into three sections: PEDOT:PSS-based perovskite solar cells, PTAA-based perovskite-based solar cells and P3CT-X-based solar cells. Finally, the ways to realize 25% inverted perovskite solar cells are discussed.

## 2. Working Mechanisms of Perovskite Solar Cells

Figure 1 presents the device architectures of a regular-type perovskite solar cell and an inverted-type perovskite solar cell. In the regular perovskite solar cell, the n-type metal oxides are widely deposited on top of the FTO/glass substrates with a post-sintering treatment. The metal oxide layer can be TiO_2_, SnO_2_, ZnO and Al-doped ZnO, which collects the photogenerated electrons and blocks the photogenerated holes simultaneously. In the inverted perovskite solar cell, the p-type polymers and p-type metal oxides are widely fabricated on top of the ITO/glass substrates with a post-thermal annealing process at about 100 °C. The high-quality perovskite crystalline thin films can be fabricated by using the two-step spin coating method with an interdiffusion process [71,72,73], the one-step spin coating method with a washing-enhanced nucleation (WEN) process [74,75,76] and the vacuum thermal co-evaporation technique [77,78,79]. The organometal trihalide perovskite can be CH_3_NH_3_PbI_3_ (MAPbI_3_), CH(NH_2_)_2_PbI_3_ (FAPbI_3_) and Cs_x_(MA_y_FA_1−y_)_1−x_Pb(I_z_Br_1−z_)_3_, mainly due to the low absorption bandgap [80,81,82], large absorption coefficient [83,84,85], small exciton binding energy [86,87,88], long exciton (carrier) lifetime [89,90,91] and high carrier mobility [92,93,94]. The capping layers for the regular perovskite solar cell and inverted perovskite solar cell are p-type small molecules [95,96,97] and n-type small molecules [98,99,100], respectively. In other words, the p-type capping layer (n-type capping layer) has to collect the photogenerated holes (electrons) and passivate the electron-rich defects (electron-poor defects) in the top region of the perovskite crystalline thin films, as shown in Figure 2. In the regular perovskite solar cell, the use of a hole modification layer can increase the hole collection efficiency and block the photogenerated electrons from the perovskite layer [101,102,103]. In the inverted perovskite solar cells, the film quality of the ETL and the contact quality at the ETL/perovskite interface can be improved via a BCP/IPA solution treatment process, which can improve V_OC_, J_SC_ and FF simultaneously [104,105,106]. Au (Cu) and Ag (Al) metals are used as the anode electrode and cathode electrode, respectively.

The highest FF values of regular perovskite solar cells and inverted perovskite solar cells are 85% [107] and 86% [108], respectively. The high FF values can be mainly explained as due to the existence of organic dipoles in the organometal trihalide perovskites, which has been demonstrated theoretically and experimentally [109,110]. The photoinduced organic dipolar alignment suggests that the electron flow and hole flow are spatially separated in the perovskite crystal and thereby eliminate the defect-mediated carrier recombination after the exciton self-dissociation. It is noted that the photogenerated hole transportation is better than the photogenerated electron transportation in the bifacial inverted perovskite solar cells, which results in the higher FF under a sunlight illumination from the semitransparent cathode electrode [111]. In other words, the hole mobility in the regular perovskite solar cell is higher than the electron mobility in the inverted perovskite solar cell, as shown in Figure 3. Fortunately, the carrier mobility of perovskites is related to the crystal orientation [112,113,114]. Therefore, it is possible to improve the photovoltaic performance of inverted perovskite solar cells via tuning the polycrystalline thin film to the most appropriate orientation plane in order to facilitate the collections of photogenerated carriers without the potential loss and carrier recombination. Conceptually, the molecular packing structure of the p-type polymers determines the preferred crystal orientation of the perovskite crystalline thin films, thereby dominating the photovoltaic performance of the resultant perovskite solar cells. In other words, the molecular structure and molecular packing structure of the HTL on top of the ITO/glass (FTO/glass) substrate can be used to understand the photovoltaic performance of the PEDOT:PSS-, PTAA- and P3CT-X-based inverted perovskite solar cells. The photovoltaic performance of the best PEDOT:PSS-, modified PTAA- and P3CT-X-based inverted perovskite solar cells is listed in Table 1.

## 3. PEDOT:PSS Thin-Film-Based Perovskite Solar Cells

In the first inverted perovskite solar cell, a PEDOT:PSS thin film is used as the HTL, which results in a moderate PCE of 3.9%. The PEDOT:PSS (1:6 wt%) thin films were widely used in the poly(3-hexylthiophene): [6,6]-phenyl C_61_-butyric acid methylester (P3HT:PCBM) blended thin-film-based OPVs as the HTL mainly due to the efficient hole-collection and electron-blocking abilities. Figure 4a presents the molecular structures of PEDOT and PSS polymers. The PEDOT and PSS are p-type polymer and large-bandgap polymer (insulator), respectively. The long-chain PEDOT polymers can be doped by shot-chain PSS polymers, thereby forming the linear molecular structure, which increases the doping concentration and work function of the PEDOT chains in the PEDOT:PSS thin films [116,117,118]. In the non-modified PEDOT:PSS thin-film-based perovskite solar cells, the highest PCE is lower than 17% mainly due to the relatively low V_OC_ and FF. The PEDOT:PSS thin films are amorphous and hydrophilic surfaces, which results in the high nucleation density during the formation of perovskite crystalline thin films, as shown in Figure 4b. In other words, the grain sizes of perovskite crystalline thin films deposited on top of the PEDOT:PSS/ITO/glass samples are smaller than 500 nm, which results in the sub-micrometer-sized perovskite grains, thereby forming the carrier recombination centers in the grains to reduce the V_OC_ and FF of the resultant solar cells. The PCE of the PEDOT:PSS thin-film-based perovskite solar cells can be increased from 15% to 18% via adding p-type graphene oxide (GO) into the HTL [119]. Conceptually, the carbon-based hydrophobic additives into the HTL can increase the grain size of perovskite crystalline thin films, which can increase the V_OC_, J_SC_ and FF simultaneously [120,121,122]. However, there is a trade-off between the grain size and surface roughness in the perovskite crystalline thin film because the thickness of the solution-processed C_60_-derivatives-based ETLs is about 50 nm. The smaller grain size results in the higher defect density in the perovskite crystalline thin film, thereby reducing the V_OC_ and FF. The larger grain size results in the roughed perovskite crystalline thin film, which cannot be completely covered by a 50 nm thick ETL and thereby reduces the V_OC_ and FF.

GO contains the hydroxyl (-OH), oxo (=O) and carboxyl (-COOH) groups [123]. The hydroxylgroup can increase the surface wettability of GO thin films, thereby increasing the contact quality at the perovskite/GO-doped PEDOT:PSS interface. The oxo group can passivate the electron-poor defect in the bottom surface of the perovskite crystalline thin films. After the dehydrogenation reaction of the carboxyl group, the ester group can be the nucleation site of the perovskite thin film, thereby increasing the contact quality at the perovskite/p-type GO interface. However, the formation of hydrogen iodide (HI) molecules can result in the iodide vacancies in the bottom region of the resultant perovskite crystalline thin film. On the other hand, the sulfonic acid (-SO_3_H) groups of the PSS in the PEDOT:PSS thin film can be considered as the nucleation sites at the perovskite/PEDOT:PSS interface after the dehydrogenation reaction. In other words, the dehydrogenation reaction of the PSS polymers can form the iodide vacancies in the perovskite crystalline thin film, which can be used to explain the formation of J_SC_ hysteresis in the J-V curves of the PEDOT:PSS thin-film-based perovskite solar cells [124,125,126]. On the other hand, the metal oxides (MoO_x_, GeO_2_ and NiO_x_) are added into the PEDOT:PSS thin films in order to increase the photovoltaic performance of the PEDOT:PSS thin-film-based perovskite solar cells [127,128,129]. Their experimental results show that the addition of metal oxides into the PEDOT:PSS thin films improves the hole collection efficiency and the contact quality at the perovskite/HTL interface.

## 4. PTAA Thin-Film-Based Perovskite Solar Cells

The p-type PTAA polymer was proposed to replace the Spiro-OMeTAD small molecule in the regular perovskite solar cells as the HTL and capping layer mainly due to higher glass transition and melting temperature [130,131]. In the regular perovskite solar cells, the thickness of the p-type capping layer is higher than 100 nm in order to completely cover the roughed perovskite crystalline thin film. To increase the hole mobility, dopants are widely added into the HTL. However, the dopants in the HTL also resulted in the shorter lifespan of the resultant perovskite solar cells mainly due to the diffusion of dopants into the active layer. Up to now, the highest PCE values of the PTAA-based regular perovskite solar cells and the Spiro-OMeTAD-based regular perovskite solar cells are 22.1% [132] and 25.6% [69], respectively. In recent years, the record high PCE values of regular perovskite solar cells were achieved by using the doped Spiro-OMeTAD thin films as the HTL. In other words, the nanometer-sized PTAA polymers cannot effectively passivate the surface defects at atomic scales, which results in the relatively lower V_OC_ and FF. Figure 5a presents the molecular structure and energy diagram of a PTAA polymer.

In the inverted perovskite solar cells, the micrometer-sized grains of the perovskite thin film can be formed on top of the hydrophobic PTAA thin film [133], which can reduce the surface defect density of the resultant perovskite thin film. When a p-doped PTAA thin film is used as the HTL, the PCE of the inverted perovskite solar cells can be increased to 17.5% [31]. However, the PCE was still lower than 20%, mainly due to the potential loss in the thick PTAA thin film. In recent years, it was found that an ultrathin face-on PTAA can be used to modify the ITO thin film, which results in higher FF and V_OC_ values. To form pinhole-free closelypacked perovskite thin film on top of the hydrophobic PTAA thin film, a two-step solvent treatment process [134] and a p-type MoO_3_ dopant [135] can be used to increase the surface wettability, thereby increasing the PCE to be higher than 20%. On the other hand, the used CuCrO_2_:PTAA inorganic-organic composite thin film increases the photovoltaic performance of the resultant perovskite solar cells mainly due to the improved hole mobility of the HTL [136].

In the best inverted perovskite solar cell, the PEAI small molecules are used to modify the surfaces of the ultrathin PTAA polymers and perovskite thin film simultaneously [70], as shown in Figure 5b. As an interlayer in between the perovskite crystalline thin film and the face-on PTAA polymers, the phenyl group of the PEAI can lie on the face-on PTAA polymers due to the π-π stacking, thereby forming the upward ethylammonium iodide group, which can be considered as the nucleation site of the perovskite thin film. As an interlayer in between the PCBM thin film and the perovskite crystalline thin film, the downward ethylammonium iodide group can passivate the iodide vacancy and organic cation vacancy, thereby forming an upward phenyl group, which can facilitate the molecular packing structure of the PCBM thin film via the π-π contact. In other words, the electron mobilities of the perovskite crystalline thin film and PCBM thin film can be simultaneously increased when the PEAI small molecules are used to modify the surface of the perovskite crystalline thin film, which can be used to explain the high V_OC_ and FF.

## 5. P3CT-X Thin-Film-Based Perovskite Solar Cells

The P3CT-Na polymer was proposed to be an alternative material to the HTL in the inverted perovskite solar cells, which resulted in a high PCE of 16.6% [33]. The P3CT-Na polymer is synthesized by mixing P3CT and NaOH in a water solution via the substitution from hydrogen anion to sodium anion. In the first P3CT-Na-based inverted perovskite solar cell, the PCE is limited to be lower than 17% mainly due to the relatively low FF and V_OC_. Figure 6 presents the molecular structure and energy diagram of a P3CT-Na polymer. P3CT is a hydrophobic polymer, which cannot be effectively dissolved in a water solution at room temperatures. The concentration of the used P3CT-Na/water solution is about 0.15 wt%, which shows that the P3CT-Na polymers can be partially dissolved in water solution due to the hydrophilicity of the Na sites. When the Na cation is replaced by K, Rb, Cs or CH_3_NH_3_ cation, the PCE of the P3CT-X-based inverted perovskite can be increased to be higher than 20% [34,137]. The main concept is that the larger cation size can minimize the formation of sub-micrometer-sized P3CT-X aggregates, thereby forming edge-on P3CT-X polymers on top of the ITO/glass substrate. The molecular structure of P3CT shows that the P3CT-X aggregates have hydrophobic surfaces due to the face-on packing structure. Figure 7 presents the edge-on P3CT-X and face-on P3CT-X on top of the ITO thin film. It is noted that the hydrophobic face-on P3CT-Na aggregates can be effectively removed from the solution by using the double-filtering process, thereby forming edge-on P3CT-Na polymers on top of the ITO/glass substrate, which increases the PCE of the resultant inverted perovskite solar cells to be higher than 20% mainly due to the relatively high J_SC_ values [35,36,37]. The higher J_SC_ value might be originated from the better crystallinity of the perovskite crystalline thin film and the better contact quality at the perovskite/P3CT-Na interface, which results in the higher exciton generation and the higher exciton dissociation (hole collection), respectively. On the other hand, the addition of graphdiyne into the P3CT-K thin film increases the photovoltaic performance of the resultant perovskite solar cells mainly due to the better surface wettability of the HTL, which improves the homogenous coverage and reduces grain boundaries of the perovskite thin film [138].

When the P3CT-X polymer is used to replace the PEDOT:PSS polymer, the J_SC_ hysteresis characteristic almost disappears in the J-V curves of the inverted perovskite solar cells. In other words, the use of edge-on P3CT-X polymers can decrease the formation of iodide vacancies in the inner region of the perovskite grains. Conceptually, the hydrophilic cations in the upper side and bottom side of the edge-on P3CT-X polymers can connect with the perovskite crystalline thin film and ITO thin film, respectively. The formation of edge-on P3CT-Na polymers is related to the surface chemical compositions (oxygen defect density) of the ITO thin films [139]. The experimental results show that the formation of edge-on P3CT-Na polymers on top of the Sn-rich ITO thin film is better, which results in a hysteresis-free and highly stable inverted perovskite solar cell. The extremely low J_SC_ hysteresis characteristic in the J-V curves shows that the upward Na sites of the edge-on P3CT-Na polymers are used to replace the organic cations as the nucleation sites of the perovskite crystalline thin films, thereby minimizing the formation of iodide vacancies.

## 6. Understanding of Highly-Efficient Inverted Perovskite Solar Cells

The formation of closelypacked perovskite crystalline thin films plays an important role in realizing the highlyefficient perovskite solar cells. The WEN process has been widely used to form the smooth and high-quality perovskite crystalline thin films on top of the various hydrophilic or hydrophobic substrates because the used antisolvents can balance the nucleation and crystal growth rates. However, the nucleation and crystal growth of the perovskite crystalline thin film are also related to the surface properties of the substrates. In 2015, it was found that the grain sizes of the perovskite crystalline thin films can be increased by decreasing the surface wettability of the substrates [140]. Therefore, the PTAA and poly[*N*,*N*′-bis(4-butylphenyl)-*N*,*N*′-bis(phenyl)-benzi (poly-TPD) thin films are widely used as the HTL of the inverted perovskite solar cells [141,142,143,144]. To completely cover the surface of the roughed ITO thin film, the thickness values of the conjugated polymer thin films are larger than 50 nm. Therefore, the additional dopants have to be used in order to increase the hole mobility of the thick conjugated polymer thin films. Besides, the micrometer-sized perovskite grains result in the relatively roughed surface, which can be used to explain why the thicker C_60_/PCBM bilayer or ZnO/PCBM bilayer is used as the ETL to cover the perovskite crystalline thin film. The device architecture of a thick HTL-based inverted perovskite solar cell is plotted in Figure 8. In the inverted perovskite solar cells, the photogenerated holes can be collected at the perovskite/HTL interface. Then, the photogenerated electrons must diffuse to the ELT/perovskite interface, which influences the generation efficiency of photocurrents. In other words, the hole mobility of HTL, the electron mobility of perovskite thin film and the electron mobility of ETL can significantly influence the carrier collection efficiency and carrier recombination rate simultaneously. The highest PCE of the thick HTL-based inverted perovskite solar cells is limited to be lower than 22% mainly due to the relatively low FF which is about 80%. In general, the nanosecond time-resolved photoluminescence decaying curves show that the photogenerated hole collection efficiency at the perovskite/HTL interface is high when the PEDOT:PSS, PTAA and P3CT-X thin films are used as the HTL [25,145,146], which means that the contact quality at the perovskite/HTL interface and hole mobility of the used HTL are both high. In other words, the limited PCE of the inverted perovskite solar cells is mainly due to the potential loss and carrier recombination, which can be used to explain the relatively low V_OC_ and FF.

In recent years, it was found that the ultrathin PTAA polymers can be used to modify the surface of a roughed ITO thin film as the anode electrode, which can result in the high PCE of 18.11% [147]. The PCE of the ultrathin-PTAA-based inverted perovskite solar cells can be increased to be higher than 21% when the surface of the hydrophobic PTAA polymers is modified by using the bipolar organic molecules, such as PEAI and 3-(1-pyridinio)-i-propanesulfonate (PPS) [148,149]. The PCE increases from 18% to 21% with the decrease in the surface wettability of the ultrathin PTAA polymers, which is mainly due to the increases in the FF and V_OC_. However, the increased FF and V_OC_ cannot be completely explained as being due to the small grains of the perovskite crystalline thin films because the surface defect density is proportional to the grain size. The higher deep-level defect density results in the lower FF and V_OC_ due to the non-radiative carrier recombination in the perovskite crystalline thin film [150]. The higher shallow-level defect density results in the higher V_OC_ because the shallow-level defects can be considered as the dopants of the perovskite crystalline thin film [151]. Besides, the smaller grains can result in a smoother perovskite crystalline thin film, which can be completely covered with a 50 nm thick PCBM thin film. When the grain size of perovskite crystalline thin films is larger, the thickness of the used ETL must be thicker in order to form a perfect PCBM/perovskite planar heterojunction. The thicker ETL results in the higher electron recombination, thereby decreasing the FF and V_OC_. In other words, the FF and V_OC_ values are limited due to the trade-off between the grain size and surface roughness of the perovskite crystalline thin film.

In 2015, the edge-on P3CT-Na polymer was used to modify the surface of the ITO thin film as the anode electrode of the inverted perovskite solar cell [33]. The optimized thickness of the P3CT-Na polymers is about 4 nm. Compared with the face-on PTAA polymers, the surface of edge-on P3CT-Na polymers is more wettable because the upward Na sites are hydrophilic. Besides, the upward Na sites of the P3CT-Na polymers can be considered as the nucleation sites during the formation of the perovskite crystalline thin film. Figure 9 presents the atomic-force microscopic images of an ITO/glass sample and a P3CT-Na/ITO/glass sample. In the ITO/glass sample, the size of islands ranges from 200 nm to 500 nm. It is noted that the deposition of the P3CT-Na polymers does not influence the surface morphology of the ITO thin film, which means that the ultrathin P3CT-Na polymer is formed. In the MAPbI_3_/P3CT-Na/ITO/glass sample, the layered surface morphology shows that the MAPbI_3_ perovskite particles are sub-micrometer-sized single crystals, as shown in Figure 10. The grain size of the MAPbI_3_ crystalline thin film is similar to the island size of the ITO thin film, which means that the MAPbI_3_ single-crystalline grains grow on top of the edge-on P3CT-Na-modified ITO islands. The crystal orientation of the MAPbI_3_ crystalline thin film deposited on top of the P3CT-Na/ITO/glass is mainly along the (110) direction [152], which is consistent with the assumption that the upward Na sites of the P3CT-Na polymers are the nucleation sites during the formation of the MAPbI_3_ single-crystalline grains.

## 7. Roles of ETL in Inverted Perovskite Solar Cells

To realize the highly efficient inverted perovskite solar cells, the photogenerated electrons in the light-absorbing layer must be collected effectively by the ETL. C_60_- and C_60_-derivatives-based thin films are widely used as the ETL of the inverted perovskite solar cells. In general, the ETL is an electron collection layer, a hole blocking layer and a passivation layer, as shown in Figure 11a,c. Without the use of a capping layer (ETL), the surface defects of the perovskite crystal can trap the photogenerated electrons and holes, as shown in Figure 11b. When the surface defects are passivated by the capping layer (ETL), the delocalized electrons and delocalized holes become free carriers, thereby generating the photocurrents. In other words, the C_60_ and C_60_ derivatives can passivate the surface defects of the perovskite crystalline thin films when they are used as the ETL of the inverted perovskite solar cells. Figure 11d presents the molecular structures of C_60_, PCBM and ICBA. When the C_60_ molecules are used as the ETL, the surface electron-rich defects of the perovskite crystalline thin films can be passivated due to the negative chargeability [153]. The negative chargeability of ICBA is better than that of C_60_ due to the higher electron affinity and non-spherical symmetric structure [154], which can be used to explain the improved photovoltaic performance of the inverted perovskite solar cells when the C_60_ is replaced by the C_70_ as the ETL [155]. When the PCBM molecules are used as the ETL, the surface electron-rich and electron-poor defects of the perovskite crystalline thin films can be passivated by the spherical fullerene and the oxygen of the functional group, respectively [29]. Besides, the molecular structure also dominates the formation of ordered molecular packing structure, which highly influences the electron mobility of the resultant ETL. Conceptually, the electron mobility of C_60_ thin films (ICBA thin films) is higher than that of ICBA thin films (PCBM thin films) due to the higher symmetry. In other words, there is a trade-off between the surface defect passivation and the formation of ordered molecular structure when the C_60_ derivatives are used as the ETL. On the other hand, the non-fullerene electron acceptor was used as an interlayer in between the ETL and perovskite crystalline thin film, which resulted in a high PCE of 22.09% mainly due to the reduced surface defects and improved carrier transport [156].

## 8. Challenges and Future Directions in Inverted Perovskite Solar Cells

Through the understanding of the PEDOT:PSS, PTAA and P3CT-Na-based inverted perovskite solar cells, the main challenges and the possible future directions are discussed in the following subsections.

### 8.1. PEDOT:PSS-Based Inverted Perovskite Solar Cells

The PEDOT chains in the PEDOT:PSS thin films are p-type heavily doped conductive polymers, which can effectively collect the photogenerated holes from the perovskite thin films without the additional potential loss. However, the FF and V_OC_ values of the PEDOT:PSS-based inverted perovskite solar cells are widely lower than 80% and 0.95 V, respectively. The main reason is the formation of small perovskite grains on top of the hydrophilic PEDOT:PSS thin film, which forms high-density defects in the perovskite crystalline thin film, thereby resulting in the non-radiative carrier recombination. A post-solvent annealing process can be used to increase the grain size of the MAPbI_3_ thin films from 250 nm to 1000 nm, which increases the J_SC_ and FF of the resultant solar cells simultaneously [157]. However, the J_SC_ hysteresis characteristic in the J-V curves can still be observed, which means the existence of halide vacancies inside the crystalline grains (point defects) in the bottom region of the perovskite thin film. Conceptually, the hydrogen cations in the sulfonic acid (-SO_3_H) groups can be substituted by the sodium cations with the addition of NaOH into the PEDOT:PSS/water solution [38], which results in the stable PSS polymers. In other words, the use of a stable PEDOT:PSS thin film with the dehydrogenation reaction can reduce the formation of halide vacancies in the perovskite crystalline thin film, which might improve the photovoltaic performance of the inverted perovskite solar cells. Besides, it is predicted that the dehydrogenation reaction of the PSS polymers in the PEDOT:PSS thin films can be performed by adding the organic halides, alkali halides or alkali hydroxide into the PEDOT:PSS/water solution. On the other hand, the doping concentration of PEDOT polymers can be largely increased by using the hydrogenosulfate as the dopant [158], which might increase the V_OC_ of the resultant perovskite solar cells.

### 8.2. PTAA-Based Inverted Perovskite Solar Cells

In the best inverted perovskite solar cell, the ultrathin PTAA polymer is modified with the PEAI bipolar organic molecules, thereby forming the bridge between the PTAA-modified ITO thin film and the perovskite crystalline thin film, which results in the high PCE of 21.58% [70]. When the surface of the perovskite crystalline thin film is modified with the PEAI molecules, the PCE increases from 21.58% to 23.72% [70]. On the other hand, the PCE of the PPS-doped perovskite solar cells increases from 20.0% to 21.7% when the PPS molecule is used as the chemical bridge [149]. The PPS dopants might mainly distribute in the top region of the perovskite thin film and thereby passivate the surface defects [149]. Conceptually, the oxygens in the sulfonate acid group of the PPS molecules can passivate the halide vacancies or the interfacial organic cations at the grain boundaries of the perovskite crystalline thin film. In other words, the improved photovoltaic performance of the PTAA-based inverted perovskite solar cells is mainly due to the vacancy reduction in the bottom region and the defect passivation in the top region of the perovskite crystalline thin films. The photovoltaic performance of the best regular and inverted perovskite solar cells is listed in Table 2. Compared with the best regular perovskite solar cell, the lower PCE of the best inverted perovskite solar cell is due to the lower J_SC_. In other words, it is possible to increase the PCE of the PTAA-based inverted perovskite solar cells to be higher than 25% by using the α-FAPbI_3_ crystalline thin film as the light-absorbing layer. However, the formation of a stable FAPbI_3_ crystalline thin film on top of the bipolar organic-molecule-modified ITO thin film will play the important role. It is noted that the grain sizes of the perovskite thin film deposited on top of the mesoporous TiO_2_/compact TiO_2_/FTO/glass substrate can be 1000 nm, which is larger than the grain size of the perovskite crystalline thin film deposited on top of the PEAI-modified ITO/glass substrate. In other words, the grain size of the FAPbI_3_ thin film deposited on top of the PEAI-modified ITO/glass substrate must be larger than the island size of the ITO thin film which ranges from 200 nm to 400 nm (see Figure 9). Fortunately, the substrate-induced small grain of the MAPbI_3_ crystalline thin film can be merged to be larger than 1000 nm via the formation of MA-C_60_-MA cations with the addition of C_60_molecules into the MAPbI_3_ precursor solution [159]. It can be predicted that the C_60_-doped FA_x_MA_1−x_PbI_3_ thin films can also form merged grains via the formation of C_60_-MA-C_60_molecules at the grain boundaries. On the other hand, the PCE of the PTAA-based inverted perovskite solar cells is proportional to the molecular weight of the used PTAA [160,161]. However, there is a trade-off between the solubility and molecular weight of polymers [162], which might limit the highest molecular weight of the used PTAA polymers in the inverted perovskite solar cells.

### 8.3. P3CT-X-Based Inverted Perovskite Solar Cells

The P3CT-X polymers can be considered as the chemical bridges between the ITO thin film and perovskite crystalline thin film via the hydrophilic Na sites, which triggers the formation of edge-on P3CT-X polymers. X can be hydrogen ion, alkali metal cation or organic cation. Conceptually, the best candidate of the downward X site is a hydrogen ion because the carboxyl group can connect with the oxygen defect of the ITO thin film after hydrogenation reaction. When the upward X sites of the P3CT-X polymers are hydrogen ions, the HI molecules will be formed during the formation of FAPbI_3_ crystalline thin film on top of the P3CT-X-modified ITO thin film via the hydrogenation reaction, which can result in the formation of iodide vacancies. Ideally, the best candidate for the upward X site is a FA cation which can be considered as the nucleation site during the formation of FAPbI_3_ crystalline thin film on top of the P3CT-X-modified ITO thin film, as shown in Figure 12a. In other words, the best P3CT-X polymer is an up-down asymmetric polymer, which can be a perfect p-type molecular bridge between the ITO thin film and FAPbI_3_ crystalline thin film, as shown in Figure 12b. However, the surface oxygen defect density of the ITO thin film must be related to the spacing between adjacent downward carboxyl groups. The d-spacing value of the FAPbI_3_ crystal along the (110) direction is larger than the spacing between adjacent upward carboxyl groups. To reduce the mismatch between the d-spacing of the perovskite crystal and the spacing between adjacent upward carboxyl groups of the P3CT polymer, a FAPbCl_3_ crystal of a FAPbBr_3_ crystal can be inserted as the buffer layer due to the shorter lattice constant.

### 8.4. An Ideal Polymer-MTL-Based Inverted Perovskite Solar Cell

Figure 13 presents the ideal polymer-based inverted perovskite solar cell. The edge-on or face-on polymers have to form an ultrathin hole modification layer (HML) on top of the roughened ITO thin film. Then, the solution-processed perovskite crystalline thin film can be grown on top of the ultrathin conjugated polymer-modified ITO thin film, which forms single crystalline perovskite grains (see Figure 10), thereby resulting in the extremely high V_OC_ and FF (see Table 1 and Table 2). In other words, the formation of molecular connection between the perovskite and HML (see Figure 12) results in the intrinsically high photogenerated hole collection efficiency. To increase the J_SC_, the optical bandgap of the light-absorbing layer used in the highly efficient inverted perovskite solar cells must be decreased by using the α−FAPbI_3_ crystalline thin film [69]. It is noted that grain size of perovskite crystalline thin films used in the highly efficient perovskite solar cells ranges from 300 nm to 2000 nm, which indicates that the photogenerated electrons can be collected effectively when the surface defects of the single crystalline perovskite grains are passivated by the capping layer (small molecular-based ETL). Besides, the thickness of the perovskite crystalline thin film is less than 600 nm due to the high absorption coefficients in the visible to near-infraredwavelength range. According to Table 2, it is predicated that the PCE of the polymer-HTL-based inverted perovskite solar cell can be increased from 23.32% to 25.47% by increasing the J_SC_ value from 24.13 mA/cm^2^ to 26.35 mA/cm^2^.

## 9. Conclusions

In summary, we have reviewed the three main p-type polymers (PEDOT:PSS, PTAA and P3CT-X) used in the inverted perovskite solar cells as the HTL or HML. In the PEDOT:PSS thin-film-based perovskite solar cells, it is predicted that the surface defects and point defects of the perovskite crystalline thin films can be reduced by adding p-type hydrophobic conjugated small molecules and by replacing the hydrogen cations of the PSS polymers with alkali metal ions, respectively, which can increase the V_OC_ and FF of the resultant solar cells. In the ultrathin face-on PTAA polymers-based perovskite solar cells, the contact qualities at the perovskite/HML interface and the ETL/perovskite interface can be improved by using the bipolar organic molecules, which increases the V_OC_, J_SC_ and FF simultaneously. In other words, it is worthwhile to develop new bipolar organic molecules to replace PEAI and PPS molecules as the chemical bridges at the perovskite/HML interface and the ETL/perovskite interface. In the ultrathin edge-on P3CT-X-polymer-based perovskite solar cells, it is predicted that the perovskite crystalline thin film and the ITO thin film can be perfectly connected with the up-down asymmetric P3CT-X polymers. Besides, the substrate-induced sub-micrometer-sized FA_x_MA_1−x_PbI_3_ grains can be merged to form micrometer-sized grains via the formation of the MA-C_60_-MA cations with the addition of C_60_ molecules into the perovskite precursor solution, which can reduce the potential loss in the light-absorbing layer, thereby increasing the V_OC_ of the resulting solar cells.

## Figures and Tables

**Figure 1 polymers-14-00823-f001:**
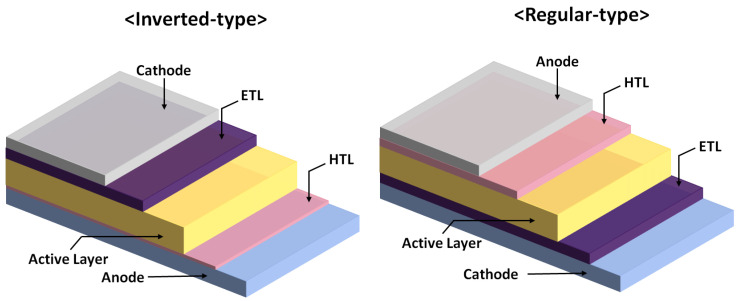
Device architectures of an inverted perovskite solar cell and a regular perovskite solar cell. ETL and HTL denote electron transport layer and hole transport layer, respectively.

**Figure 2 polymers-14-00823-f002:**
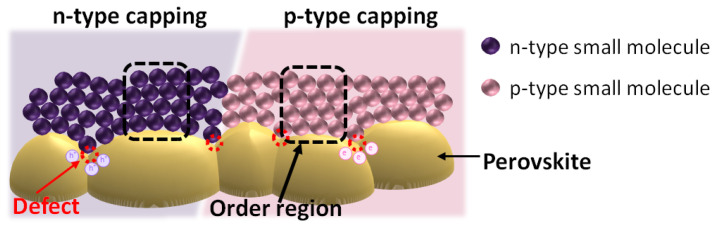
ETL/perovskite and HTL/perovskite interfaces.

**Figure 3 polymers-14-00823-f003:**
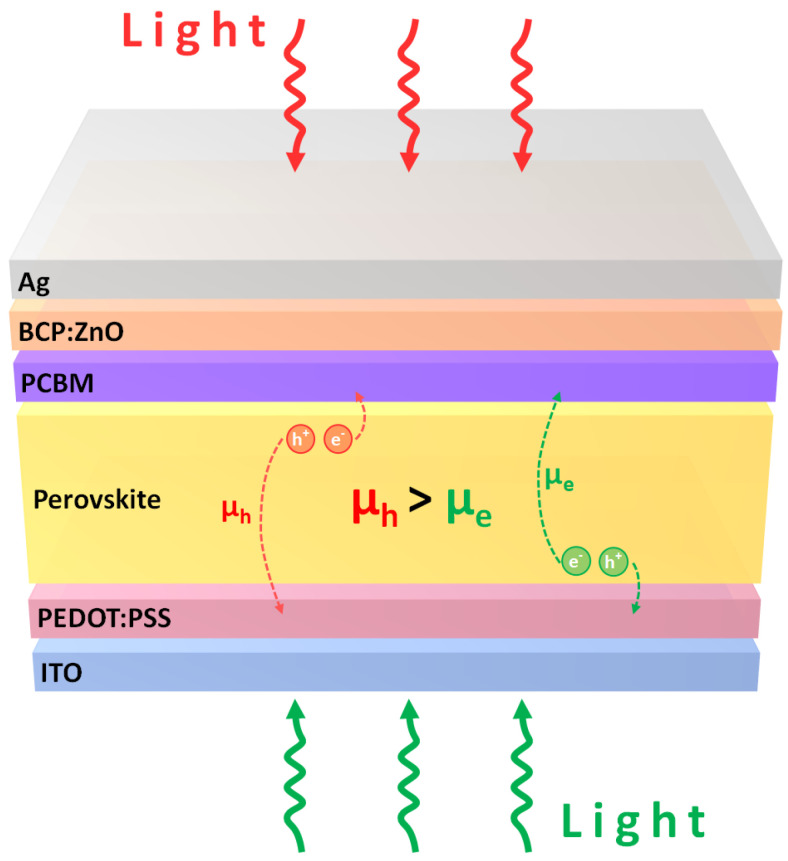
Electron and hole transportations of a bifacial perovskite solar cell.

**Figure 4 polymers-14-00823-f004:**
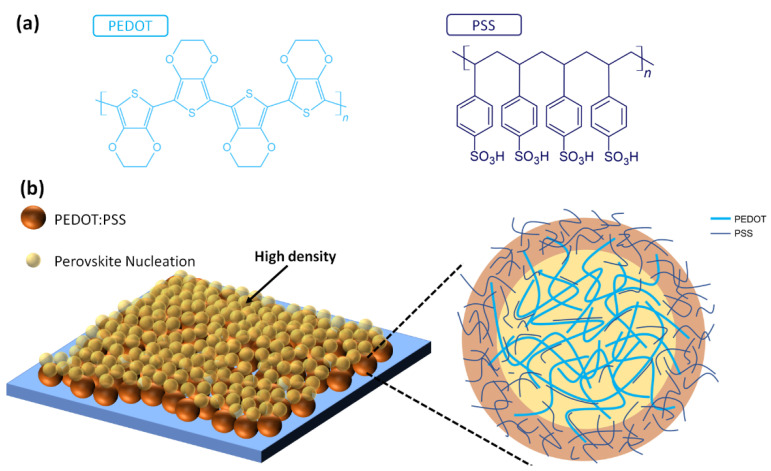
(**a**) Molecular structures of PEDOT and PSS polymers. (**b**) High-density perovskite nucleation points on top of the hydrophilic PEDOT:PSS thin film.

**Figure 5 polymers-14-00823-f005:**
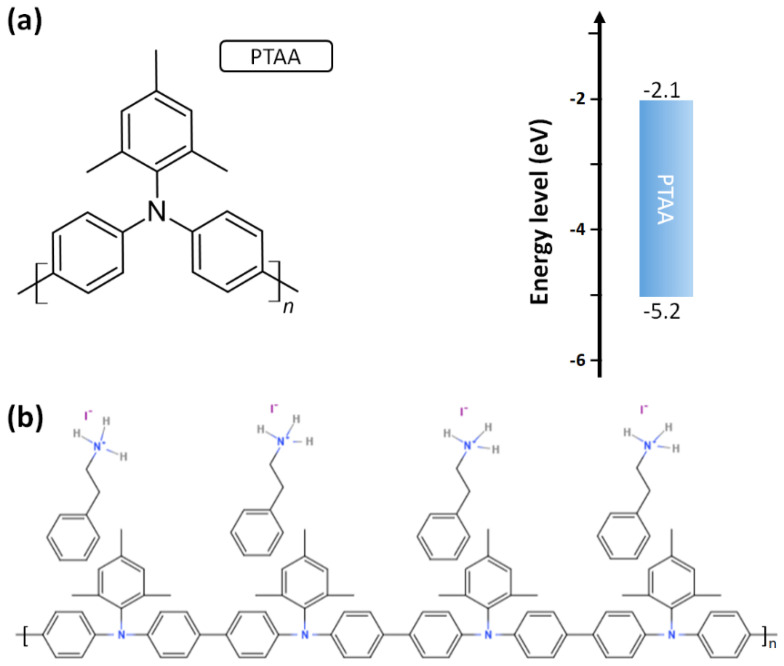
(**a**) Molecular structure and energy diagram of PTAA. (**b**) PEAI-modified PTAA.

**Figure 6 polymers-14-00823-f006:**
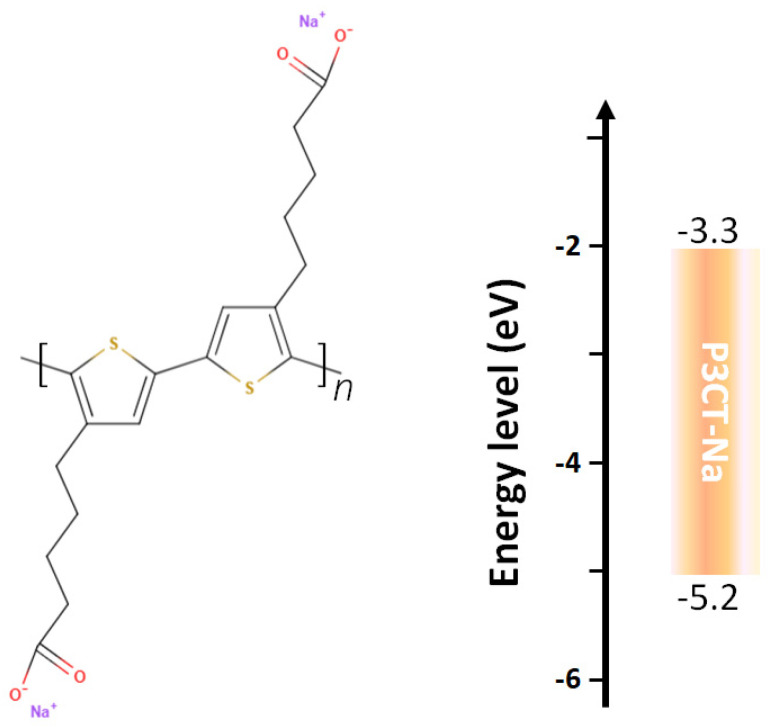
Molecular structure and energy diagram of P3CT-Na.

**Figure 7 polymers-14-00823-f007:**
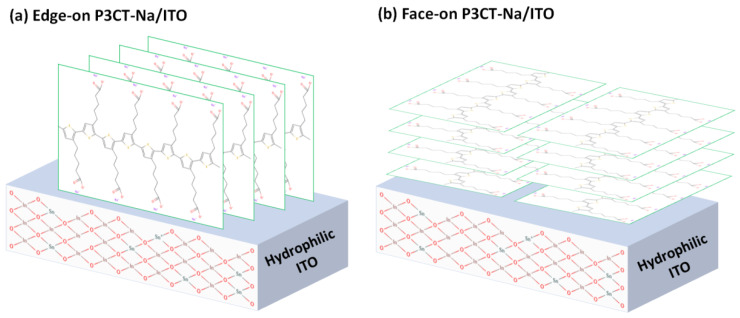
P3CT-Na polymers on top of the hydrophilic ITO thin film. (**a**) Edge-on. (**b**) Face-on.

**Figure 8 polymers-14-00823-f008:**
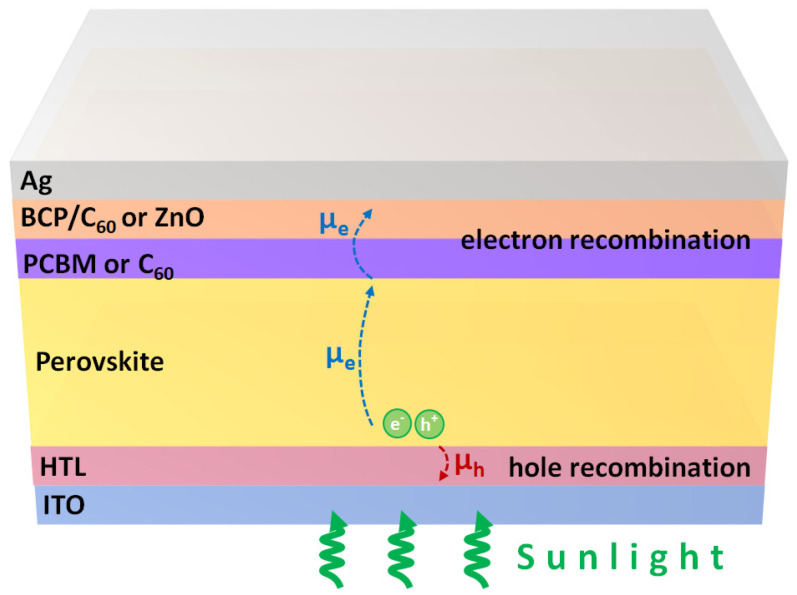
The main carrier transportations in the thick HTL/ETL-based inverted perovskite solar cell. HTL denotes hole transport layer. The ETL can be BCP/C_60_/PCBM or ZnO/C_60_.

**Figure 9 polymers-14-00823-f009:**
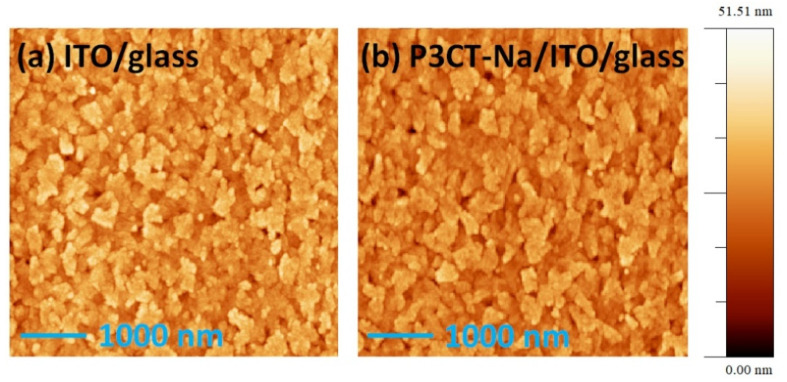
AFM images. (**a**) ITO/glass; (**b**) P3CT-Na/ITO/glass.

**Figure 10 polymers-14-00823-f010:**
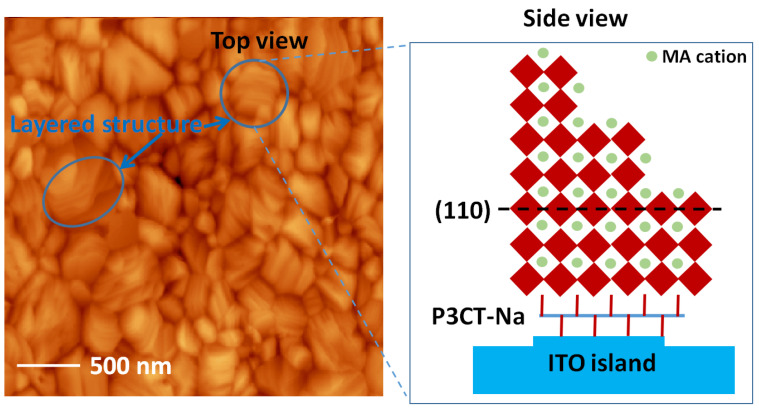
AFM image of a MAPbI_3_/P3CT-Na/ITO/glass substrate and the schematic side view of the MAPbI_3_/P3CT-Na/ITO trilayer structure.

**Figure 11 polymers-14-00823-f011:**
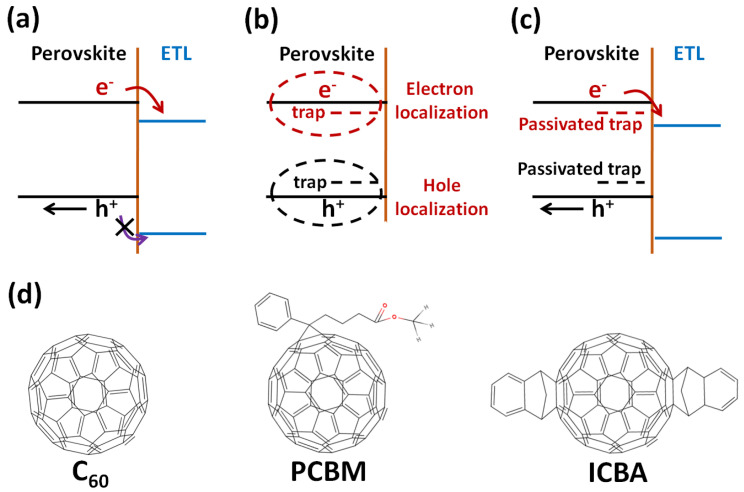
Energy diagrams and molecular structures. (**a**) Electron collection and hole blocking. (**b**) Trap-induced carrier localization. (**c**) Surface defect passivation. (**d**) C_60_, PCBM and ICBA.

**Figure 12 polymers-14-00823-f012:**
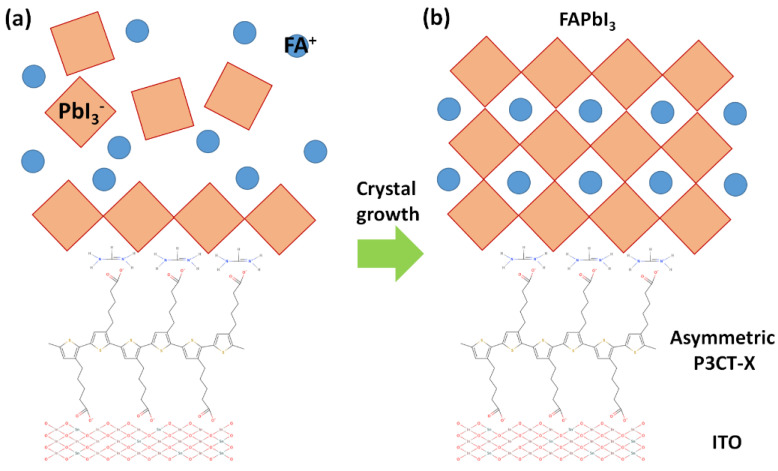
Formation ofsingle crystalline FAPbI_3_ perovskite on top of the asymmetric P3CT-X-modified ITO thin film. (**a**) Nucleation. (**b**) Crystal growth.

**Figure 13 polymers-14-00823-f013:**
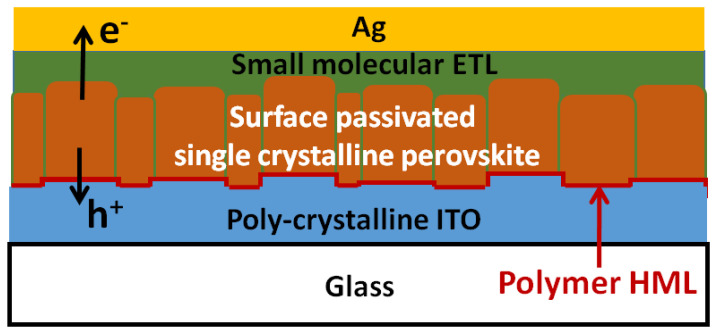
The device structure of an ideal polymer-HML-based inverted perovskite solar cell.

**Table 1 polymers-14-00823-t001:** Photovoltaic performance of the best PEDOT:PSS-, modified PTAA- and P3CT-X-based inverted perovskite solar cells under one sun illumination (AM 1.5 G, 100 W/cm^2^).

P-Type Polymer	Perovskite	Grain Size and Thickness of Perovskite (nm)	V_OC_ (V)	J_SC_ (mA/cm^2^)	FF (%)	PCE (%)	Ref.
PEDOT:PSS	MAPbI_3_	1500/470	1.060	23.10	86.0	21.05	[108]
Modified PTAA	(MAFA)Pb(ICl)_3_	350/550	1.155	24.13	83.7	23.32	[70]
P3CT-X	(CsMAFA)Pb(IBr)_3_	300/400	1.120	22.78	83.6	21.33	[115]

**Table 2 polymers-14-00823-t002:** Photovoltaic performance of the best regular and inverted perovskite solar cells under one sun illumination (AM 1.5 G, 100 W/cm^2^).

Structure Type	Perovskite	Grain Size and Thickness of Perovskite (nm)	V_OC_ (V)	J_SC_ (mA/cm^2^)	FF (%)	PCE (%)	Ref.
Regular	FAPbI_3_	2000/none	1.189	26.35	81.7	25.59	[69]
Inverted	(MAFA)Pb(ICl)_3_	350/550	1.155	24.13	83.7	23.32	[70]

## Data Availability

Not applicable.

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
