# Peer review of "Understanding the PEDOT:PSS, PTAA and P3CT-X Hole-Transport-Layer-Based Inverted Perovskite Solar Cells"

_polymers, 2022, doi:10.3390/polym14040823_

Round 1
Reviewer 1 Report
The article does not include an in-depth analysis of the influence of the polymer chemical structure on the photovoltaic parameters of inverted perovskite solar cells. Tables for all presently tested polymers along with photovoltaic parameters should be presented and discussed.
There is no information in the abstract about the type of polymers used in inverted perovskite solar cells.
There is no presentation of the 3 groups of polymers other than those mentioned, which makes the article not very innovative on the basis of the current reviews on this subject.
If I understand correctly, the presented Figures are from other papers. There is no information on where the Figures was taken from and are the publisher's consent provided? Should read: Copyright permission from .....
The article should be re-examined in terms of the proposed title and should present the latest trends in the use of polymers in inverted perovskite solar cells. As the authors know very well, this subject develops very rapidly and every week there are new reports in the field of analysis and interpretation of phenomena occurring in these subject.
Author Response
We thank the reviewer for the comments and suggestions. Our responses are listed in the attached file.

Reviewer 2 Report
In this review manuscript the authors carried out an analysis of the state of art of inverted perovskite solar cells. They cover a revision of three main p-type hole transporting material layers and correlate their composition, structural and hydrophilic properties, with the nucleation of perovskite crystals and cell performance parameters. After a deep reading I consider the topic is worth enough to trigger the interests of polymer journal audience. However, major revision would be required to improve the submitted review scientifically
- As this review focus on the effect of HTM composition, manufacture and chemical properties on PS layer structural properties and final device performance. I encourage to replace the title by a more specific one (such as understanding the role of HTL in polymers based inverted perovskite solar cells).
- I consider that a comprehensive analysis and discussion of the role of electron transporting polymers in inverted solar cell should be included.
- I suggest to include a summary table illustrating the most representative works for different polymeric HTL, and connecting their performance with their structural properties (crystal size distribution, roughness) and with processing conditions (HTL composition, dopants).
- I should like to have seen more detailed figures, including illustration of oriented growth mechanism for different dopant or HTL mechanism, or (with journal permission) reproduction of some of the most representative works.
- A deeper study which includes correlation of time transient or frequency domain analysis with cell performance and structural properties and connection with physical elements that govern the electronic and ionic dynamics in perovskite solar cell needs to be included.
Author Response

(The authors gave the same response as above.)

Reviewer 3 Report
(Review Article): Polymers
Title: Progress in understanding of polymer based inverted perovskite solar cells
Comments:
The author reviewed the recent progress of polymer-based inverted perovskite solar cells. It mainly focuses on understanding the polymer-based hole transport layer (HTL) in perovskite solar cells.
They covered the following contents:
- The PEDOT: PSS, PTAA, and P3CT-X polymers HTL and its impact on inverted perovskite solar cell performance.
- It also covered the challenges and future direction for polymer-based inverted perovskite solar cells.
The manuscript is well prepared and could be improved.
My comments are following.
- The title does not define the contents inside and needs to modify. It may contain the term hole transport layer as the manuscript is all about polymer HTL.
- Abstract also need to modify a bit as it does not contain clear information about the contents inside. The line “polymer-based inverted perovskite solar cells” is a bit confusing and not clear about the use of polymer (Like polymer-based charge transport layer or polymer doped perovskite solar cells).
- In sections 3-5, the author describes the polymer HTLs, and most of the examples contain the pristine HTLs. It could be good to add a few examples about the dopped materials in these HTLs and how that affects the device performance parameters.
- In challenges section, contains the general information and does not describe current challenges/issues with polymer HTLs and could be improved a bit.
Author Response

(The authors gave the same response as above.)

Round 2
Reviewer 1 Report
I see progress in the paper and I proposed accepted paper as is.
Reviewer 2 Report
The authors have revised the manuscript according to the suggestion. And the work itself is of interest to people working on the relevant field and can be acceptable in Polymers. However, I still have some concern that needs to be addressed before publication. See details bellow.
-As I mentioned in previous report I consider that the role of ETL in this inverted solar cell should be treated with more detail in a separated section. It would improve the overall quality of the review.
-The summary table illustrating the most representative works should be extended, including device performance, cell features, structural details (grain size, layer thickness) and covering the most representative works of the state of art for such devices.
-The authors have to devote more efforts in improving the quality of figures. Including the more representative works and a scheme which correlates the perovskite structural parameters with cell performance.
Author Response
We thank the reviewer for the useful suggestions. Our replies are listed in the attached file.

Reviewer 3 Report
The authors have addressed the issues; I am pleased to commend this revised version acceptable for publication in Polymers.
Author Response
We thank the reviewer for reading our revised manuscript.
Round 3
Reviewer 2 Report
I am satisfied with the responses provided by the authors. The manuscript in my estimation is suitable for publication.